# LtGAPR1 Is a Novel Secreted Effector from *Lasiodiplodia theobromae* That Interacts with NbPsQ2 to Negatively Regulate Infection

**DOI:** 10.3390/jof9020188

**Published:** 2023-01-31

**Authors:** Caiping Huang, Junbo Peng, Wei Zhang, Thilini Chethana, Xuncheng Wang, Hui Wang, Jiye Yan

**Affiliations:** 1Beijing Key Laboratory of Environment Friendly Management on Fruit Diseases and Pests in North China, Institute of Plant Protection, Beijing Academy of Agriculture and Forestry Sciences, Beijing 100097, China; 2Center of Excellence in Fungal Research, Mae Fah Luang University, Chiang Rai 57100, Thailand; 3School of Science, Mae Fah Luang University, Chiang Rai 57100, Thailand

**Keywords:** *Lasiodiplodia theobromae*, plant immunity, effector, PsbQ2

## Abstract

The effector proteins secreted by a pathogen not only promote the virulence and infection of the pathogen but also trigger plant defense response. *Lasiodiplodia theobromae* secretes many effectors that modulate and hijack grape processes to colonize host cells, but the underlying mechanisms remain unclear. Herein, we report LtGAPR1, which has been proven to be a secreted protein. In our study, LtGAPR1 played a negative role in virulence. By co-immunoprecipitation, 23 kDa oxygen-evolving enhancer 2 (NbPsbQ2) was identified as a host target of LtGAPR1. The overexpression of NbPsbQ2 in *Nicotiana benthamiana* reduced susceptibility to *L. theobromae*, and the silencing of NbPsbQ2 enhanced *L. theobromae* infection. LtGAPR1 and NbPsbQ2 were confirmed to interact with each other. Transiently, expressed LtGAPR1 activated reactive oxygen species (ROS) production in *N. benthamiana* leaves. However, in NbPsbQ2-silenced leaves, ROS production was impaired. Overall, our report revealed that LtGAPR1 promotes ROS accumulation by interacting with NbPsbQ2, thereby triggering plant defenses that negatively regulate infection.

## 1. Introduction

Unlike animals, plants lack mobile immune cells and an adaptive immune system [1]. However, plants are protected against microbes by physical barriers, such as strong and thick epidermis cells, and by the innate immunity of each cell [2]. Plants possess a two-tiered immune system comprising pathogen-associated molecular pattern (PAMP)-triggered immunity (PTI) and effector-triggered immunity (ETI) [3].

Grapevine canker disease caused by *Lasiodiplodia theobromae* (*L. theobromae*) has become a significant fruit disease, causing considerable yield loss in the grape industry. However, an efficient method for controlling this disease has not been found since the current understanding of the pathogenic mechanism is still limited. Comparative genomics and transcriptomic analysis revealed that many genes related to cell wall degradation, nutrient uptake, secondary metabolism, and membrane transport have gene family expansion between *L. theobromae*, *Botryosphaeria dothidea*, and *Neofusicoccum parvum* [4]. Nearly 500 candidate-secreted effector proteins have been predicted. The secretory endopolygalacturonase protein, LtEPG1, acts as an elicitor for *L. theobromae* virulence [5]. A putative effector, LtCSEP1, from *L. theobromae*, inhibits BAX-triggered cell death and suppresses immune responses [6]. These studies provide novel insights into the pathogenic mechanism and provide new ideas for treating this disease.

LtGAPR1 is a putative effector that is upregulated during *L. theobromae* infection of grapes [4]. It contains an SCP GAPR-1 domain (Golgi-associated plant pathogenesis-related protein 1). Proteins that contain the SCP GAPR-1 domain have been reported in some opportunistic pathogenic fungi, such as *L. theobromae*, *Diplodia seriata*, and *D. corticola* [4,7]. LtGAPR1 is a secreted PR-1 (pathogenesis-related 1)-like protein found in a variety of eukaryotes, including fungi, insects, and mammals, and it has been implicated in both plant and animal immune systems [8,9,10]. In some of the classifications, PR-1 proteins belong to the CAP protein superfamily, including the Cysteine-rich secretory proteins in vertebrates, Antigen 5 in insects, and PR-1 in plants [11,12]. A PR-1-like protein, Fpr1, from *Fusarium oxysporum*, is required for full virulence on a mammalian host but dispensable for virulence on plants [9]. However, how LtGAPR1 modulates disease symptom development remains largely unknown.

Oxygen-evolving enhancer proteins (OEEs) have three subunits: OEE1 (PsbO), OEE2 (PsbP), and OEE3 (PsbQ). They are nuclear-encoded chloroplast proteins that bind photosystem II (PSII) on the luminal side of the thylakoid [13]. PsbQ is involved in the stabilization of the complex that catalyzes the photolysis of water, the first step in the non-cyclic electron transport of photosynthesis [14]. In this process, two molecules of water are oxidized to one molecule of oxygen and four protons, with the generation of four electrons that travel through the photosynthetic electron transport chain. Tomato PsbQ has been reported to interact with the effector HopN1 of *Pseudomonas syringae* to suppress plant innate immune responses [15]. The PsbQ protein stabilizes the functional binding of the PsbP protein to photosystem II in higher plants [16]. However, PsbP also functions in many biotic stresses. For example, the specific protein (SP) of rice stripe virus (RSV) interacts with PsbP to enhance virus symptoms [17]. The βC1 protein of a geminivirus betasatellite interacts with PsbP and changes PsbP-mediated antiviral defense in plants [18]. Therefore, PsbQ may also play a role in biotic stress because of its high similarity to PsbP. The function of PsbQ in biotic and abiotic stress needs to be further explored.

In this study, we revealed that an *L. theobromae* effector, LtGAPR1, was highly induced during early infection. RNAi or the overexpression of this gene confirmed that LtGAPR1 negatively regulated the infection process. Furthermore, co-immunoprecipitation (Co-IP) and bimolecular fluorescence complementation assay (BiFC) were used to validate the interaction between LtGAPR1 and NbPsbQ2. Transient expression of LtGAPR1 led to ROS production by DAB staining; however, in NbPsbQ2-silenced plants, ROS production decreased in both LtGAPR1-expressing and *L. theobromae*-infected plants. This means that LtGAPR1 interacted with NbPsbQ2 to manipulate ROS production to promote plant immunity against *L. theobromae*.

## 2. Results

### 2.1. LtGAPR1 Is a Secreted Protein with an SCP-GAPR1-like Domain

LtGAPR1 was predicted as an effector by bioinformatic analysis previously, for which the gene ID is 56011904. Full-length amino acid sequence analyses using the Pfam programs showed that LtGAPR1 contains a signal peptide (SP) in the N-terminal and an SCP-GAPR1-like domain (Figure 1A). The GAPR1-like domain has been reported to share sequence and structural similarities with the PR-1 protein in plants (Appendix A), which supports the notion that the GAPR1 protein may represent an evolutionary link between plant and mammalian immune systems [12]. The GAPR1 gene is upregulated in mammalian cells with cancer; therefore, it may be important for pathogenesis [12]. Interestingly, *LtGAPR1* is also unregulated in the early stage of *L. theobromae* infection (Appendix A). The method for identifying secreted proteins has been well established based on invertase secretion for yeast cells growing on media with sucrose or raffinose as the sole carbon source [19,20]. Furthermore, pSUC2:LtGAPR1, the positive control pSUC2:Avr1b, and the negative control pSUC2:Mg87 were separately transformed into the yeast strain YTK12, which was defective in invertase secretion. All strains grew on YPDA media, while yeast carrying pSUC2 grew on CMD-W media. Only the positive control grew on YPRAA media (Figure 1B). LtGAPR1 grew on YPRAA media similar to the positive control, which means that the SP of LtGAPR1 functions perfectly. LtGAPR1 is a secreted protein.

### 2.2. LtGAPR1 Is Not a Necessary Growth Factor but a Negative Virulence Factor of Vitis vinifera

To estimate the contribution of LtGAPR1 to virulence, this gene was silenced or overexpressed via polyethylene-glycol-mediated protoplast transformation. The positive transformants were confirmed by qRT-PCR analysis (Appendix A). The in vitro growth of LtGAPR1-RNAi2, LtGAPR1-RNAi5, LtGAPR1-RNAi7, and LtGAPR1-RNAi9 was measured on PDA plates after 24 and 48 h. The diameter showed no difference between the control and RNAi strains (Figure 2A), indicating that silencing LtGAPR1 did not affect vegetative growth. However, the pathogenicity test on detached grapevine shoots showed that LtGAPR1-OE11, LtGAPR1-OE12, LtGAPR1-OE13, and LtGAPR1-OE14 had decreased disease symptoms (Figure 2B), and the silenced transformants had increased disease symptoms (Figure 2C). Thus, LtGAPR1 plays a negative role in the pathogenicity of *L. theobromae*.

### 2.3. LtGAPR1 Interacts with NbPsQ2

To further explore the potential mechanism by which LtGAPR1 negatively regulates the virulence of *L. theobromae*, immunoprecipitation–mass spectrometry was performed, and LtGAPR1-GFP, as bait, was expressed in the leaves of *Nicotiana benthamiana*. A total of 20 candidates were found. Oxygen-evolving enhancer protein 2-3, also named PsbQ2, is one of these candidates. PsbQ2 is a nuclear-encoded chloroplast protein that binds to the periphery of photosystem II (PSII) on the luminal side of the thylakoid [13].

To further determine whether LtGAPR1 interacts with NbPsbQ2, GFP-tagged LtGAPR1 without its signal peptide was co-expressed with Flag-tagged NbPsbQ2 in *N. benthamiana*. All proteins were successfully expressed in the input samples, and LtGAPR1-GFP was detected in NbPsbQ2-Flag immunoprecipitation but not in Flag immunoprecipitation, suggesting that LtGAPR1 was associated with NbPsbQ2 in planta (Figure 3A).

To supply more cogent evidence about the interaction between LtGAPR1 and NbPsbQ2, we used a bimolecular fluorescence complementation assay (BiFC). The YFP signal was present at the periphery of the chloroplast in LtGAPR1-CeYFP, and NbPsbQ2-NeYFP co-expressed leaves showed that the two proteins interact at the periphery of the chloroplast (Figure 3B). These results support the physical interaction of LtGAPR1 and NbPsbQ2.

### 2.4. Overexpression of NbPsbQ2 Reduces Susceptibility to L. theobromae Infection, and Silencing NbPsbQ2 Enhances the Susceptibility of L. theobromae

Because LtGAPR1 was associated with NbPsbQ2, it was necessary to determine the role of NbPsbQ2 in plant resistance to *L. theobromae*. Two days after transient expression of NbPsbQ2-GFP or GFP and three weeks after silencing NbPsbQ2 by TRV: NbPsbQ2, *N. benthamiana* was inoculated with *L. theobromae*. The silenced leaves were confirmed by RT-PCR (Appendix A). TRV: NbPsbQ2 had significantly larger lesions than TRV plants (Figure 4A); furthermore, NbPsbQ2-GFP had smaller lesions than GFP plants (Figure 4B). This suggests that NbPsbQ2 negatively regulates plant resistance to *L. theobromae*.

### 2.5. LtGAPR1 Triggers ROS Production but Is Impaired in NbPsbQ2-Silenced Plants

Transient expression of LtGAPR1 induced ROS production (Figure 5B). Since PsbQ2 has been reported to be involved in ROS production and LtGAPR1 interacted with PsbQ2, to determine whether LtGAPR1 induced ROS production is related to the targeting of PsbQ2, we inoculated *N. benthamina* leaves (WT and NtPsbQ2-silenced) with *L. theobromae*. In NbPsbQ2-silenced leaves, ROS production decreased more than in the WT leaves (Figure 5A). Additionally, in transiently expressing LtGAPR1-GFP plants, ROS production was impaired in NbPsbQ2-silenced leaves compared to the GFP-expressed leaves, as shown by the DAB test (Figure 5B). Therefore, we conclude that LtGAPR1 interacted with NbPsQ2 to promote ROS production, acting against the virulence of *L. theobromae*.

## 3. Materials and Methods

### 3.1. Strain, Plant Materials, and Culture Conditions

The *L. theobromae* strain CSS-01s, the wild-type, overexpressed transformants, and the silenced transformants were cultured on a complete medium (6 g yeast extract, 3 g casein acid hydrolysate, 3 g casein enzymatic hydrolysate, and 10 g sucrose per liter) at 25 °C. *Agrobacterium tumefaciens* strain GV3101 and *Escherichia coli* strain DH5α were cultured on Luria–Bertani medium (5 g yeast extract, 10 g tryptone, 10 g NaCl per liter, and 16 g agar was added per liter for plates). *Nicotiana benthamiana* plants were grown from seeds in soil with a 14 h/10 h light/dark period at 25 °C. Healthy green shoots of *Vitis vinifera* cv. “Summer Black” were obtained from Xiangyi field vineyard in Shunyi, Beijing, China.

For functional validation of the putative signal peptide, the predicted signal peptide sequence of LtGAPR1 was cloned to pSUC2 and transformed into yeast strain YHK12 using the Frozen-EZ yeast transformation II kit (Zymo Research, Irvine, CA, USA). YPDA medium, CMD-W medium (6.7 g yeast nitrogen base without amino acids, 0.7 g tryptophan dropout supplement, 20 g sucrose, 1 g glucose, and 20 g agar per liter), and YPRAA medium (10 g yeast extract, 20 g peptone, 20 g raffinose, and 2 mg antimycin A per liter) were used to plate the transformants. The signal peptides of Avr1b from *Phytophthora sojae* and the 25 amino acids of the non-secreted Mg87 protein from *Magnaporthe oryzae* were used as positive and negative controls, respectively.

### 3.2. Co-Immunoprecipitation Assay and Mass Spectrometry

Leaves of 4-week-old *N. benthamiana* plants were agroinfiltrated with pCAMBIA1300-LtGAPR1-GFP, and two days after infiltration, the leaves were collected and sent to Novogene to perform the immunoprecipitation assay and mass spectrometry assay.

Leaves of 4-week-old *N. benthamiana* plants were agroinfiltrated with pCAMBIA1300-LtGAPR1-GFP and pCAMBIA1300-flag-NbPsbQ2. *Agrobacterium* with the P19 silencing suppressor was added in a 1:1 ratio at a final OD_600_ of 0.5 for each construct. Two days after agroinfiltration, the leaves were frozen in liquid nitrogen and ground into a fine powder using a mortar and pestle. Protein was extracted using RIPA lysis buffer by mixing 1 g of leaf tissue with 2 mL of lysis buffer. The samples were centrifuged at 4 °C for 10 min at 14,000× *g*, and the supernatant was transferred to a new tube. A 100 μL protein extract was mixed with 10 µL normal mouse IgG and Flag primary antibody at a dilution ratio of 1:1000. The mix was incubated at 4 °C for 2 h with rotation. After preincubation, 40 µL protein A/G PLUS agarose was added to the sample and incubated overnight at 4 °C. The samples were then centrifuged at 3000 rpm for 5 min at 4 °C. The supernatant was carefully discarded, and 1 mL of IP lysis buffer was added to wash the beads, which were then centrifuged three times at 3000 rpm for 5 min at 4 °C. Finally, the precipitate in a 60 μL volume of washing buffer was mixed with 20 µL 4× loading buffer, heated at 95 °C for 5 min, and centrifuged for 5 min at maximum speed. The proteins were then loaded on a gel or stored at −20 °C.

### 3.3. Bimolecular Fluorescence Complementation Assay

Bimolecular fluorescence complementation (BiFC) assays were performed using pCAMBIA-NeYFP and pCAMBIA-CeYFP vectors. The ORF of LtGAPR1 was cloned into pCAMBIA-CeYFP. The ORF of NbPsbQ2 was cloned into pCAMBIA-NeYFP. *Agrobacterium tumefaciens* strain GV3101 carrying the indicated construct was co-infiltrated into 4-week-old *N. benthamiana* leaves. Fluorescence was detected at 48 h post infiltration (hpi) using a confocal laser scanning microscope.

### 3.4. Virus-Induced Gene Silencing

VIGS was performed in *N. benthamiana* to silence NbPsbQ2 using the pTRV vectors pTRV1 and pTRV2:NbPsbQ2. pTRV2:NbPDS1 was used as a positive control. *Agrobacterium tumefaciens* containing different constructs was inoculated into LB medium with 10 mM MES, 10 mM MgCI_2_, and 200 µM acetosyringone overnight at 28 °C. The cultures were harvested, prepared in infiltration buffer as a 1:1 mixture (pTRV1/pTRV2; final OD_600_ = 1), and incubated at room temperature for 3 h. The *A. tumefaciens* inoculant was then infiltrated into the first pair of true leaves of 2-week-old soil-grown plants using a needleless syringe. The infiltrated plants were covered overnight with a transparent dome in a growth room and then moved into a growth chamber for 2–3 weeks until the positive control, pTRV2:PDS1, showed a phenotype. The successfully silenced leaves were confirmed by RT-PCR.

### 3.5. Agrobacterium-Mediated Transient Expression

For agroinfiltration experiments, 4- or 5-week-old plants were used. Cultures of *A. tumefaciens* (strain GV3101) carrying plasmids were harvested by centrifugation, resuspended in infiltration medium (10 mM MES, 10 mM MgCI_2_, and 200 µM acetosyringone; pH 5.5) to a final concentration of 0.5 at 600 nm (OD_600_), and infiltrated into the abaxial side of the leaf using a syringe without a needle. After agroinfiltration for 48 h, the leaves were observed under confocal microscopy.

### 3.6. Grapevine Inoculation Test of Overexpressed and RNAi Transformants of the LtGAPR1 Gene

The *L. theobromae* wild-type strain, CSS-01s, was cultured on complete medium (6 g yeast extract, 3 g casein acid hydrolysate, 3 g casein enzymatic hydrolyzate, and 10 g sucrose per liter) at 25 °C for 2–5 days, and the mycelial plugs (4 mm in diameter) were cut with a cork borer. The inoculation test was performed on detached *Vitis vinifera cv*. Summer Black green shoots, following [21], or on the back of *N. benthamiana* leaves that had been stabbed with sterilized insect needles. The mycelial plugs were placed on the wound with the hyphae face down, and each *N. benthamiana* leaf was set as a replicate with a total of 30 replicates. A blank PDA medium was used as a control. The inoculated tissues were cultured in a greenhouse at 25/18 °C in the light/dark with 12 h light/dark alternation and 80% relative humidity.

The LtGAPR1 open reading frame (ORF) was amplified with primers (Appendix A) and subcloned into a modified pKSNTP vector. Subsequently, the fusion constructs referred to as pKSNTP-LtGAPR1 were transformed into the *L. theobromae* protoplast using the polyethylene-glycol-mediated transformation method described by [4]. The resultant transformants were screened against neomycin resistance and confirmed by qRT-PCR analysis.

For RNAi transformation, we amplified the sense fragment with primers (Appendix A) and ligated both fragments into the *pRTN* vector in the given order. Subsequently, fusion vector *pRTN:*
*LtGAPR1* was transformed into the *L. theobromae* protoplast similar to overexpression vector transformation; the protocols used for pathogenicity tests of RNAi transformants were also similar to those of the overexpressed transformants.

### 3.7. Quantification of DAB Staining

Four hours after fungal inoculation, *N. benthamiana* was stained in a freshly prepared 1 mg mL^−1^ solution of 3,3′-diaminobenzidine (DAB, Sigma-Aldrich D-8001) in 8 mM HCl (pH 3.8). Chlorophyll was removed by submerging the leaves into a solution of ethanol/lactic acid/glycerol (3:1:1 (vol/vol/vol)) at 60 °C and keeping them overnight at room temperature on water-soaked filter paper. At least six biological replicates from each specimen were mounted on slides and observed with a Nikon SMZ1500 microscope under a bright field.

## 4. Discussion

The ascomycete fungus, *L. theobromae*, is an economically important and destructive fungal pathogen [6,22,23]. However, our understanding of the molecular basis of *L. theobromae* pathogenicity is still limited. In recent years, hundreds of effector proteins predicted from *L. theobromae* have been found, but the details of how complex defense/counter-defense cross-talk between effectors and grapevines need to be further explored.

LtGAPR1 is identified as an *L. theobromae* effector and is highly aligned with the allergen V5/Tpx-1-related protein in humans. Based on the functional validation of LtGAPR1, it can be secreted into the extracellular environment. However, when quantifying LtGAPR1 gene expression at different temperatures, the gene was downregulated under higher temperatures (Appendix A), while pathogenicity became enhanced at higher temperatures. In a further virulence study, LtGAPR1 enhanced grapevine defense against *L. theobromae* at higher temperatures. These results suggest that LtGAPR1 may play a negative role in infection. As *L. theobromae* is a weak opportunistic pathogen, it may deploy its effector, manipulating its pathogenicity. Previous research has shown that effector proteins secreted by a pathogen not only promote virulence and infection of the pathogen but also trigger a plant defense response [24]. The overexpression of *Magnaporthe oryzae* systemic defense trigger 1 (MoSDT1) confers improved rice blast resistance in rice [24]. Interestingly, transient expression of LtGAPR1 induced ROS production by DAB staining (Figure 5A,B). Therefore, LtGAPR1 negatively regulated virulence, perhaps by promoting ROS production.

To explore the deep mechanism, using LtGAPR1-GFP as a probe, the chloroplast protein, NbPsbQ2, was identified as a target in *N. benthamiana*. Co-IP and BiFC experiments showed that LtGAPR1 and NbPsbQ2 interacted with each other. NbPsbQ2, a protein of OEC of PSII, has previously been reported to be related to stress conditions, and silenced plants showed normal photosynthetic activity under normal light conditions, but the efficiency changed under low-light conditions [25,26]. Moreover, few studies have shown that protein variation in OECs is associated with viral infections. Peanut green mosaic virus infection leads to changes in polypeptides on PSII particles [27]. The silencing of PsbO enhances virus replication in *Nicotiana* plants [28]. A reduction in the PsbO transcript level in wheat reduces the sporulation of *Puccinia striiformis* [29]. Although the specific mechanisms underlying these phenomena have been elusive, a previous report showed that PAMP-induced PTI inhibits carbon fixation in the chloroplast. This creates excess excitation energy in plants under illumination, resulting in ROS generation in the chloroplast [15]. This suggests that the photosynthesis apparatus participates in a general defense response against different types of pathogens and can manipulate this response to its advantage by slowing down the generation of harmful ROS.

In our study, transiently expressing LtGAPR1-GFP caused active ROS production, while in NbPsbQ2-silenced leaves, DAB staining showed impaired ROS production. Interestingly, when expressing LtGAPR1-GFP in NbPsbQ2-silencing leaves, there was not strong ROS production as in the wild-type, indicating that LtGPAR1 interacts with NbPsQ2 for active ROS production to further enhance plant immunity.

In summary, our results show that the effector, LtGAPR1, from *L. theobromae*, targeted NbPsbQ2, a chloroplast protein controlling the ROS-mediated defense response, to enhance plant immunity against *L. theobromae*. This study explained opportunistic pathogens’ strategy in invading grapevine, in which it deployed its effector to retain pathogenicity for infecting plants (Figure 6).

## Figures and Tables

**Figure 1 jof-09-00188-f001:**
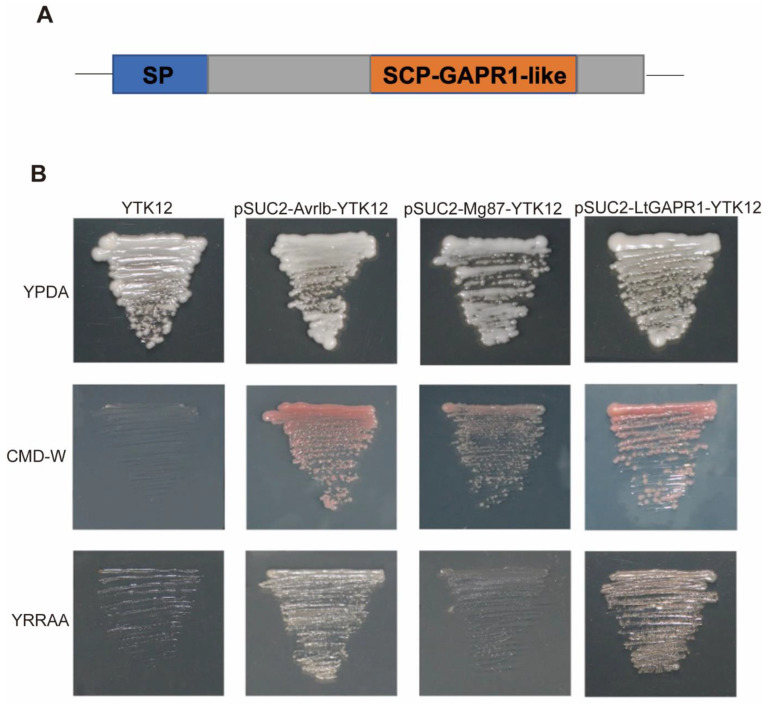
Conserved domain and functional validation of the predicted signal peptide of the *L. theobromae*, LtGAPR1. (**A**) Conserved domain of LtGAPR1 containing a signal peptide (SP) and SCP-GAPR1 domain. (**B**) Encoding sequence of the predicted signal peptide of LtGAPR1 fused upstream of the invertase gene sequence. Yeast YTK12 transformed with the fusion vectors was grown on YPDA, CMD, and YPRAA media. All yeast strains grew on YPDA. Transformants carrying the pSUC2 fusion vector grew on CMD-W. Only transformants whose invertase was secreted to the extracellular environment grew on YPRAA. YTK12: parental strain; pSUC2-Avrlb-YTK12: positive control; pSUC2-Mg37-YTK12: negative control; pSUC2-x-YTK12: allergen transformants.

**Figure 2 jof-09-00188-f002:**
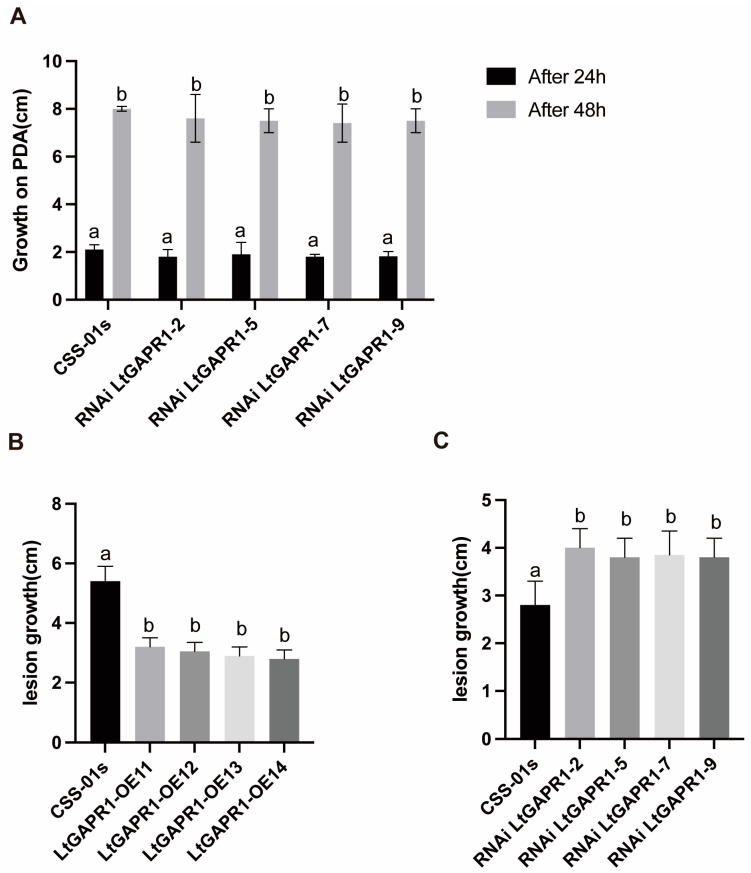
LtGAPR1 negatively regulated the virulence of *L. theobromae* but did not affect its vegetative growth. (**A**) Vegetative growth of the wild-type *L. theobromae* strain, CSS-01s, and several RNAi-LtGAPR1 strains on PDA at 24 and 48 h. (**B**) Statistical analysis of lesion length caused by overexpressed LtGAPR1 transformants. One-year-old grapevine shoots were inoculated using mycelial plugs (5 mm in diameter) and kept in a chamber under constant humidity and temperature. The inoculated grapevines were photographed at 3 days post inoculation (dpi). Bar = 1 cm. (**C**) Statistical analysis of lesion length caused by the silenced transformants of LtGAPR1. The means and standard errors were calculated from five replicates. One-way ANOVA was performed for statistical analysis. Isolates that do not share the same letter (a, b) exhibit a significant difference among the isolates.

**Figure 3 jof-09-00188-f003:**
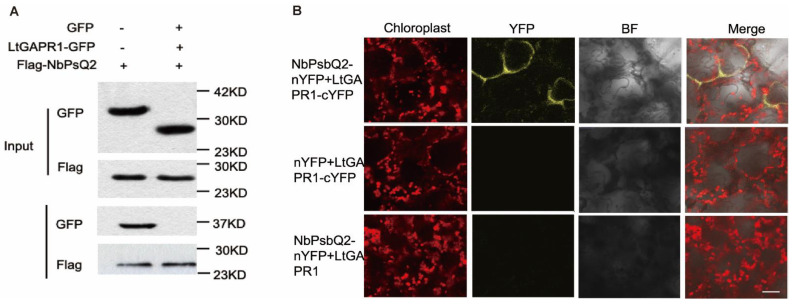
LtGAPR1 interacts with NbPsQ2. (**A**) Interaction analysis of LtGAPR1 with NbPsbQ2 via the Co−IP assay. (**B**) Interaction analysis of LtGAPR1 with the NbPsbQ2 assay via split YFP assays. *Agrobacterium tumefaciens* expressing p35S:nYFP−NbPsbQ2 and p35S:cYFP−LtGAPR1 were co−infiltrated into 4−week−old *Nicotiana benthamiana* leaves. The images were taken using a confocal laser scanning microscope at 2 days post infiltration (dpi). Bar = 50 μm.

**Figure 4 jof-09-00188-f004:**
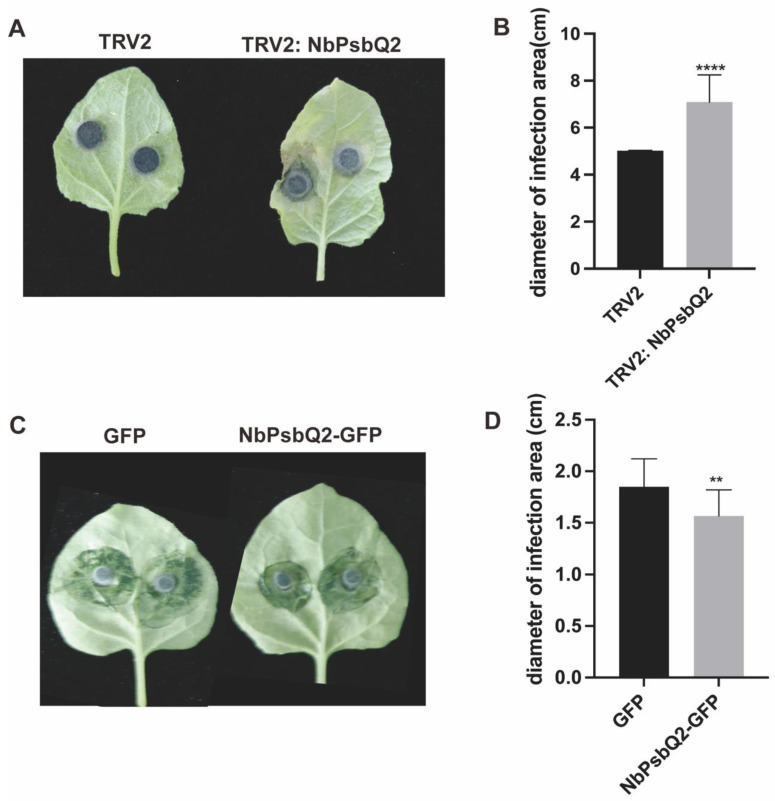
NbPsbQ2 promotes *N. benthamiana* immunity to *L. theobromae*. (**A**) Transiently expressed NbPsbQ2-GFP and GFP in *N. benthamiana* showed promotion of plant immunity to *L. theobromae*. Images were taken at 4 dpi. (**B**) Diameters of the infection area. Values are the means of 20 biological replicates. Significant differences were evaluated using a *t*-test and the least significant differences using an (LSD) test (** α = 0.01; **** α = 0.0001). (**C**) Transient expression of NbPsbQ2-GFP and GFP in *N. benthamiana* promoted plant immunity to *L. theobromae*. Images were taken at 4 dpi. (**D**) Diameters of the infection area. The values are the means of three biological replicates. Significant differences were evaluated using a *t*-test and the least significant differences using (LSD) test (** α = 0.01; **** α = 0.0001).

**Figure 5 jof-09-00188-f005:**
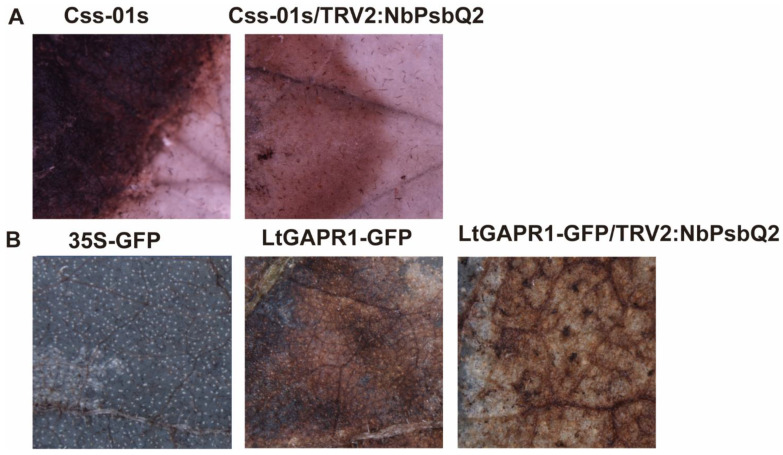
ROS production in NtPsbQ2-silenced plants. (**A**) WT and NbPsbQ2-silenced *N. benthamiana* were inoculated with *L. theobromae* mycelial plugs (5 mm in diameter) and then kept in a chamber under constant humidity and temperature for 24 h before DAB staining. (**B**) ROS production detection in transiently expressing GFP leaves, transiently expressing LtGAPR1-GFP leaves, or NbPsbQ2-silenced leaves and LtGAPR1-GFP-expressing leaves, as shown by the DAB test.

**Figure 6 jof-09-00188-f006:**
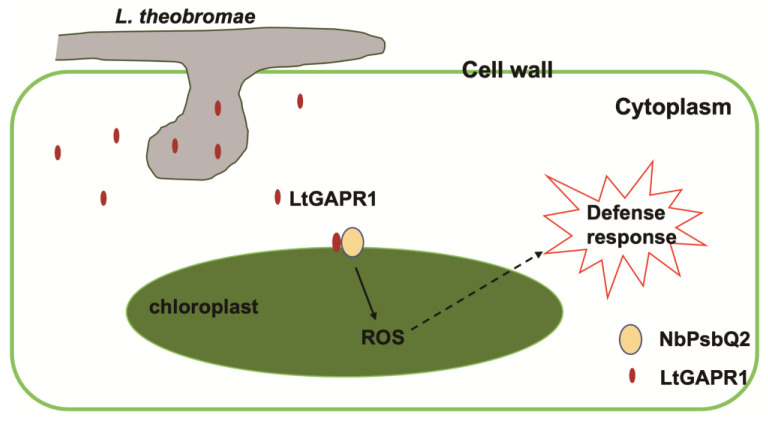
A working model illustrating how LtGAPR1 interacts with NbPsbQ2 for active ROS production to further enhance plant defense.

## Data Availability

The data presented in this study are available on request from the corresponding author.

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
