# Peer review of "LtGAPR1 Is a Novel Secreted Effector from Lasiodiplodia theobromae That Interacts with NbPsQ2 to Negatively Regulate Infection"

_jof, 2023, doi:10.3390/jof9020188_

Round 1

Reviewer 1 Report

The manuscript by Huang et al identified a novel secreted effector from Lasiodiplodia theobromae LtGAPR1, which plays a negative role in virulence by targeted NbPsQ2 from Nicotiana benthamiana. The authors first confirmed that LtGAPR1 was a secreted effector in L. theobromae, then showed that LtGAPR1 negatively regulated the virulence of L. theobromae by combining loss of function and overexpression experiments. The authors further demonstrated that NbPsQ2 from Nbenthamiana could associated with LtGAPR1 using co-immunoprecipitation, and showed thatLtGAPR1 promotes ROS accumulation by interacting with NbPsbQ2, thereby triggering plant defenses that negatively regulate infection. The experimental approaches used by the authors are appropriate and data provided are in good quality. In my opinion, this manuscript is worth to publish, but still I have some suggestions.

Minor concerns:

1.     Line 87-89, Fig 1A, it would be better to compare the structure domain of LtGAPR1 with PR-1 protein in plants and mammalian cells.

2.     Line 84, please indicate why to study LtGAPR1 and where is this protein from, and it gene ID at the beginning section of Result. 

3.     Line 136-137, why indicate PsbQ2 here? It seems that PsbQ2 was co-precipitated with LtGAPR1, please indicate it here.

4.     Line 165, in Fig 4D showed “ d of infection area (cm)”, please indicate the full name of “d”

5.     Line 175, please explain why detect ROS production here, and where is the ROS from (N. benthamina?)

Author Response

1.     Line 87-89, Fig 1A, it would be better to compare the structure domain of LtGAPR1 with PR-1 protein in plants and mammalian cells.

Thanks for your  suggestion, we have added an alignment of LtGAPR1 and PR-1 protein in Arabidopsis, human, mouse in supplemental figures. 

2.     Line 84, please indicate why to study LtGAPR1 and where is this protein from, and it gene ID at the beginning section of Result. 

We had added  the information in  Line 87. 

3.     Line 136-137, why indicate PsbQ2 here? It seems that PsbQ2 was co-precipitated with LtGAPR1, please indicate it here.

We are grateful for your keen observation, We have made the correction.(Line 147)

4.     Line 165, in Fig 4D showed “ d of infection area (cm)”, please indicate the full name of “d”

Thanks for your kindly reminder, we have changed the “d” for “diameter” for Figure 4D.

5.     Line 175, please explain why detect ROS production here, and where is the ROS from (N. benthamina?)

Thanks for your  good question,  we had added the information in Line 186-187. The ROS production were observed in the area of the leaves where LtGAPR1 expressed and L. theobromae  challenge.

Reviewer 2 Report

The author shows us a series of interesting experiments demonstrating that the effector LtGAPR1 from Lasiodiplodia theobromae targeted NbPsbQ2, a chloroplast protein, controlling the ROS-mediated defense response, to enhance plant immunity against L. theobromae. The study explained opportunistic pathogens’ strategy in invading grapevine, in which it deployed its effector to retain pathogenicity for infecting plants. It is worth mentioning that many of the materials and methods used in this study, such as Overexpression of NbPsbQ2 in Nicotiana benthamiana and Virus-induced gene silencing, are quite innovative, which can be used for further mechanistic investigation…

A couple of minor points:

What is the ROS mentioned many times in the manuscript and abstract? The full name should be indicated for the first time…

In the last paragraph of the Introduction section, the author should be briefly stated that the materials and methods used in this study correspond to the experimental results, so that the readers can better understand this study.

Line 144. “in vitro” should be italicized.

In the legend of Figure 2, the test method and significant differences should be supplemented. Besides, an overall description of the data analysis should be added to the Methods section.

Check the ordinate of Figure 4D…

Figure 6. A working model should be further beautified

Only twenty-four references are cited in this paper, which is not enough for a research article. More relevant studies can be cited that have been published to strengthen the argument of this study in the Discussion section.

Author Response

What is the ROS mentioned many times in the manuscript and abstract? The full name should be indicated for the first time…

Thanks for your good suggestion, we have revised.(Line 20)

In the last paragraph of the Introduction section, the author should be briefly stated that the materials and methods used in this study correspond to the experimental results, so that the readers can better understand this study.

Thanks for your suggestions, We had revised according your suggestion.(Line 80-84)

Line 144. “in vitro” should be italicized.

Thanks for your kindly reminder, we have changed.(Line 126)

In the legend of Figure 2, the test method and significant differences should be supplemented. Besides, an overall description of the data analysis should be added to the Methods section.

We added the information in Line 143-144

Check the ordinate of Figure 4D…

Thanks for your kindly reminder, we have changed the “d” for “diameter” for Figure 4D.

Figure 6. A working model should be further beautified

Thanks for the suggestion, we have tried our best best to beautified it.

Only twenty-four references are cited in this paper, which is not enough for a research article. More relevant studies can be cited that have been published to strengthen the argument of this study in the Discussion section.

Thanks for your professional comments, we added the references till to thirty now.

Reviewer 3 Report

This manuscript reported an L. theobromae  secreted protein LtGAPR1 promotes ROS accumulation by interacting with NbPsbQ2, thereby triggering plant defenses that negatively regulate infection. The authors  identified the secreted proteins based on invertase secretion for yeast cells. This gene was silenced or over expressed via polyethylene glycol-mediated protoplast transformation to estimate the contribution of LtGAPR1 to virulence. A bimolecular fluorescence complementation assay was used to investigated the interaction between LtGAPR1 and NbPsbQ2. The methods were reasonable and the result can supported the conclusion.  I sugguested  add the necessary references to introduction and discussion parts, e.g. in line 46 LtGAPR1 is a putative effector that is upregulated during L. theobromae infection of grapes.(ref),e.g. in line 299 "Previous research has shown that effector proteins secreted by a pathogen not only promote virulence and infection of the pathogen but also trigger a plant defense response(ref). 

Author Response

I sugguested  add the necessary references to introduction and discussion parts, e.g. in line 46 LtGAPR1 is a putative effector that is upregulated during L. theobromae infection of grapes.(ref),e.g. in line 299 "Previous research has shown that effector proteins secreted by a pathogen not only promote virulence and infection of the pathogen but also trigger a plant defense response(ref). 

Thanks for professional comments, we added the references till to thirty now.